# Pregnancy physical activity questionnaire (PPAQ): Translation and cross cultural adaption of an Arabic version

Tatiana Papazian[1,2], Nada El Osta[3,4,5], Hala Hout[2], Daisy El Chammas[2], Nour El Helou[2], Hassan Younes[2,6], Georges Abi Tayeh[7], Lydia Rabbaa Khabbaz[1]*

1 Laboratoire de pharmacologie, pharmacie clinique et contrôle de qualité des médicaments, Faculty of Pharmacy, Saint-Joseph University of Beirut, Beirut, Lebanon, 2 Department of Nutrition, Faculty of Pharmacy, Saint-Joseph University of Beirut, Beirut, Lebanon, 3 Department of Prosthodontics, Faculty of Dental Medicine, Saint-Joseph University of Beirut, Beirut, Lebanon, 4 Laboratoire de Recherche Crâniofaciale, Unité de Santé Orale, Facuty of Dental Medecine, Saint-Joseph University of Beirut, Beirut, Lebanon, 5 Centre de Recherche en Odontologie Clinique, Clermont University, University of Auvergne, EA 4847, Clermont-Ferrand, France, 6 Department of Nutrition and Health Sciences, Institut Polytechnique LaSalle Beauvais, Beauvais, France, 7 Department of Gynecology and Obstetrics, Hotel Dieu de France Hospital, Ashrafieh, Beirut, Lebanon

* lydia.khabbaz@usj.edu.lb

**Data Availability Statement:** All relevant data are within the paper and its Supporting Information files.

**Funding:** This study received no funding.

## Abstract

### Introduction

Physical activity level during pregnancy is unknown in Middle Eastern and North African countries, since no valid tools assessing it exist in Arabic. The aim of this study is to culturally adapt and translate to Arabic an internationally validated instrument, the Pregnancy Physical Activity Questionnaire (PPAQ), and to measure the physical activity of pregnant women using the adapted PPAQ, Arabic version. This tool is time-sparing, self-administered and is the only one taking into account childcare and household chores.

### Methods

After following the guidelines for translation and back-translation by certified translators, a committee composed of professionals in the field reviewed each item of the PPAQ, for its comprehensibility. This Arabic version of the PPAQ was tested on a sample of 179 pregnant Lebanese women, from different educational backgrounds, socioeconomic status and gestational ages.

### Results

Cross-cultural adaptations were applied on the newly translated PPAQ in Arabic version, thus questions referring to some types of outdoor activities were excluded from the final format. Our results reported that 51% and 1.7% of women engage respectively in light and high intensity physical activity, while 18% had a sedentary lifestyle. Occupational type of activities were significantly more performed by women having a higher education (p value 0.001), as opposed to those who attended only high school, who were physically more

**Competing interests:** The authors have declared that no competing interests exist.

**Abbreviations:** ACOG, American College of Obstetricians and Gynecologists; BMI, Body mass index; GWG, gestational weight gain; MET, Metabolic equivalent for task; PA, Physical activity; PPAQ, Pregnancy physical activity questionnaire.

active in household activities (p value 0.038). Second trimester was a period where pregnant women were active for household, caregiving (p value 0.031), whereas women in their third trimester were physically more active in occupational activities (p value 0.001). Sport-oriented activities were performed by women reporting a good physical status (p value 0.03). Age and crowding index were significantly correlated with occupational, household and caregiving activities (p values 0.004, 0.008 respectively). No significant correlations were observed with pre-gestational body mass index and the physical activity levels.

## Conclusion

A valid tool will help researchers in Arab countries identify physical activity levels of pregnant women and consequently emit specific guidelines relative to the importance and the benefits of a daily active lifestyle throughout gestation.

## Introduction

Physical activity (PA) together with proper nutrition are key contributors of maternal and fetal well-being. Numerous studies highlight the positive impact of an active lifestyle on low incidence of maternal complications [1] such as preeclampsia, gestational diabetes [2], and excess maternal weight gain [3]. The American College of Obstetricians and Gynecologists (ACOG) recommends pregnant women to engage in 30 minutes of moderate intensity exercise on daily basis [4]. However, despite the recommendations of medical instances, in some cultures, pregnant women are advised to follow a sedentary lifestyle [5]. In addition, due to religious sensitivities, exercising in public areas or local gyms are forbidden for women in some Middle-Eastern countries [6].

Reports published worldwide focus on the knowledge and the practice of PA among pregnant women and explore maternal physical and psychological factors that enhance or hinder this behavior [7–11]. Hence, with the expansion of researches on the advantages of PA on health, various questionnaires have been developed such as the Godin Leisure-Time Exercise Questionnaire [12], the Simple Physical Activity Questionnaire [13], and the Global Physical Activity Questionnaire [14], but few were appropriate for pregnant women, mainly because they did not take into account daily routine activities frequently performed by them, such as childcare and household, were long and not validated during pregnancy [15]. Compared to the other tools, The Pregnancy Physical Activity Questionnaire (PPAQ) was the only self-administered tool, with sufficient reliability in assessing total PA and vigorous activities during the last three months of gestation, preceding the survey [16]. It has been translated into different languages and used in numerous clinical studies [17–20].

Hence, a valid PA tool in Arabic is necessary to help researchers assess, monitor and evaluate PA in North African and Middle-Eastern pregnant women. However, translation alone can't guarantee the compatibility of the questionnaire, because it must undergo some cross-cultural adaptations, to ensure its validity with the local culture [21]. Hence, the aim of this research was primary to translate and cross-culturally adapt this tool into Arabic and secondly to evaluate the PA levels of healthy Middle Eastern pregnant women by applying this newly translated version of PPAQ. In addition, the associations between some maternal variables and PA levels were analyzed.

## Methods

Besides being easy, cost-effective, time sparing and noninvasive, this specific tool takes into account the time spent in 33 different activities divided as follows: 16 activities dealing with household and caregiving, 3 concerning transportation and inactivity, 9 sports related activities, and 5 occupational activities. Each participant chooses the answer which represents the closest to the amount spent on daily or weekly basis, during the preceding three months. Six possible answers are available to respondents: never, less than half an hour per day, between half and 1 hour per day, 1 and 2 hours per day, 2 and 3 hours a day, and more than 3 hours a day. In addition, two open ended questions, concerning sport type activities, allow the responder to specify types not listed previously and practiced on weekly basis. This gives the questionnaire an added value, by individualizing the evaluation of each responder. The original English version of the PPAQ takes approximately 10 minutes to be filled up. After the obtention of the written approval of its creator, Lisa Chasan-Taber, by mail, for cross-culturally translating and using this tool in a sample of Lebanese pregnant women, the study was divided into two stages: preparing the Arabic version and evaluating the PA levels of our target population.

The research team followed the guidelines depicted by Guillemin et al., which were composed of 4 phases, designed to guarantee the reliability of the translated tool [17].

### Phase 1: Translation

The translation of the original English questionnaire to Arabic was conducted by separate translators to produce several versions of the translated document. Initial translations to Arabic were performed by two nutrition bilingual graduates, who were unaware of the purpose of the study. A qualified translator whose native language was Arabic, and having a linguistic but not a scientific background, was engaged to submit a third translated version.

### Phase 2: Back-translation

Experts recommend conducting independently the same number of back-translation from the final language (Arabic) to the original English version, as done in the first phase. This helps to highlight all the ambiguities present in the translated version, and hence confirm accuracy. Two senior university students, specializing in translation and a qualified translator (not related to the first translator) were given the task to undergo the back-translation. Once again, the intentions and the objectives related to the document were not revealed to the translators, to minimize biases and subjective inputs.

### Phase 3: Committee review

A multidisciplinary team was created in our department composed of a gynecologist, a midwife, two nutritionists, two sports experts, and two graduate assistants. Their aim was to review each item in each translation separately, in order to come up with the final translated version during joint meetings. Both versions were compared for its semantic equivalence and to ensure the full comprehensibility of the questionnaire, and the instructions to fill it. A final consensus was made after discussions and synthesis.

### Phase 4: Pre-testing

For pre-testing purposes, Beaton D. et al [22] recommend testing it on a sample of 10 to 40 persons. Hence, before administrating it on a larger scale, 45 pregnant women, mostly instructors, university and hospital employees, filled the newly translated Arabic version, as part of a pilot study, to test the proper comprehensibility of the items. The research team used the same

format of the initial PPAQ concerning the font used and the sequences of questions. In addition, they were recording all aspects dealing with the questionnaire from lay-out, flow of question, time and ambiguities. A probing approach was chosen during this phase, to have a direct feedback on the clarity of the questions presented.

After completion of this pretesting, some revisions were needed. The first 3 questions (date of recruitment, last menstrual period, and predicted date of delivery) were unanimously judged unnecessary, since as mentioned before, the PPAQ was going to be used with a main questionnaire concerning the health and the nutritional status of pregnant women, and those questions were already addressed. Hence, the numbering of the questions was corrected and the final questionnaire started with 1 referring to the question 4 of the initial document. To avoid confusing the respondents, some changes were done concerning the lay-out of the questionnaire such as writing the questions with similar wording in bold characters.

After this initial phase of translation, some cultural adaptations had to be implemented too. Those concerned questions number 18 and 19 of the original version, which consisted of assessing the time spent on mowing the lawn and showering the snow off. Those activities can't be realized in Middle Eastern and Arab countries, which have hot, humid and desert climate environment. With those omissions, the number of items of our PPAQ dropped to 31. Other changes included the conversion of English measurements (gallons and pounds) mentioned in the original questionnaire into metric equivalents (liters and kilograms). We preferred mentioning "the last three months of your pregnancy", instead of last trimester, to avoid some misinterpretations (S1 File).

Our tool was ready to be administered on a larger scale, since our research team was studying the nutritional status of pregnant women and the impact of diet, exercise and other factors on maternal complications and neonatal outcomes in a cross-sectional study. The inclusion criteria were to include Lebanese women, with singleton pregnancy, Arabic literate, aged between 18 and 40 years old, healthy, not suffering from gestational diabetes, preeclampsia, or any other chronic disease affecting their health.

The minimum sample size to be included in this study was calculated initially by following the formula of Tabachnick and Fidell [23] that takes into consideration the number of explanatory variables to be include in the model: $N = 50 + 8m$ (m being the number of explanatory variables); Given that m equals to 8, a minimum of 114 women should participate in this study. They were recruited during prenatal visits, for routine gynecological check-ups, independently of their weeks of gestation. Face to face interviews permitted to collect sociodemographic data and details concerning dietary intakes. The indicators used in the first part of the questionnaire dealing with the socioeconomic status were the crowding index (deducted by the interviewer as the total number of co- residents per household, excluding the newborn infant, divided by the total number of rooms, excluding the kitchen and bathrooms), the educational level, and the work status. A crowding index higher than one suggests a household with restricted economic resources [24]. At the end of the first part of this questionnaire, the research team requested from the participants to subjectively assess their physical and nutritional wellbeing, by choosing from the list "very bad, bad, average, good and excellent", the status that currently fits them best.

Anthropometric measurements (weight before pregnancy, actual body weight, height), age, and gestational age were copied directly from the medical file, the day of the recruitment, to avoid ending up with irrelevant information. Body mass index (BMI) was then calculated as the ratio of pre-gestational weight in kilograms to the square of height in meter. It was categorized according to the World Health Organization (WHO) cut-off points (underweight <18.5, normal 18.5 to 24.9, overweight 25 to 29.9 and obese >30) [25]. The last part of the questionnaire, concerned the newly translated PPAQ and was self-filled by the participants themselves,

in the presence of the research assistants. Their role was to avoid having any missing data. A follow-up was maintained with participants thru telephone calls to compile information on delivery outcomes (weight at hospital admission, delivery complications . . .). Total GWG was then calculated by deducting weight at delivery from the initial and comparing it to the values defined by the Institute of Medicine (IOM) [26]. Field work was conducted between January and May 2018 by trained dietitians.

The study protocol was approved by the Institutional Review Board of Saint-Joseph University at Beirut, Lebanon, the Hotel-Dieu Hospital Ethics Committee (CE 624/ FPH 49) and the participating gynecologists. All subjects gave their written consent prior to participation.

## Statistical analysis

Descriptive data was calculated as the mean +/- standard deviation. Data collected was analyzed using SPSS Data Analysis version 22.0 (SPSS Inc, Chicago, IL). The alpha error was set at 0.05. Kolmogorov-Smirnov tests were used to assess the normality of the distribution of continuous variables. Univariate followed by multivariate analyses were performed to assess the factors associated with PPAQ. Spearman Correlation coefficient and Pearson Correlation coefficient were used to assess the relationship between continuous variables. Kruskal-Wallis tests and Analysis of variance (ANOVA) were performed to compare continuous variables between three or more groups. Multiple regression analysis was used and explanatory variables with a p value less than 0.200 in univariates analyses were included in the model. Variables highly correlated together were not included in the same model.

We followed the calculation guidelines described by Chasan-Taber et al. [27] to deduct the daily energy expenditure and classify the participants into different metabolic equivalent (MET) categories: sedentary ($<1.5$ METs), light ($1.5<3.0$ METs), moderate ($3.0–6.0$ METs), or vigorous ($>6.0$ METs). Briefly, each self-reported answer of each category was multiplied by its MET value, specified by the original author, to reach the total of daily energy expenditure (METs x hours/day), knowing that 1 MET is the metabolic equivalent of the energy expended at rest and equals 3.5 ml $O_2$/kg/min. The final values for each question were summed up to reach a total score that was divided by the total number of questions and then by 7 to reach the mean daily energy expenditure expressed in METs. Following those calculations, women were classified as having a sedentary, light, moderate or vigorous PA levels.

## Results

### Demographic characteristics

One hundred and seventy-nine women participated in this study, with a mean age of 29.94 ± 5.03 years old. Fifty nine percent of our sample had a university degree, with a mean crowding index of 0.73. The majority of the participants (68.2%) were in their third trimester, and the rest were equally distributed in their first and second trimester. The mean BMI calculated based on their pre-gestational weight was 22.94± 3.60 kg/m², reflecting a sample within the normal BMI range before gestation.

Women were asked to self-report their nutritional status and physical well-being. The majority of the women reported an excellent and a good physical and nutritional status. Detailed results of all the descriptive variables are presented in Table 1.

### Levels of physical activity

Mean MET value of our sample was 2.52, suggesting that women participating in this study engaged in low levels of PA. Classifications according to the MET values revealed that only

**Table 1. Descriptive variables of the sample population (N = 179).**

|  | Mean ± SD |  |  |  |
|---|---|---|---|---|
| Age | 29.94± 5.03 |  |  |  |
| Crowding Index | 0.72± 0.31 |  |  |  |
| Pre-gestational BMI[‡] | 22.94± 3.60 |  |  |  |
|  |  | N | % |  |
| **Education** |  |  |  |  |
| Elementary |  | 3 | 1.70 |  |
| Intermediate |  | 12 | 6.70 |  |
| Secondary |  | 15 | 8.40 |  |
| Technical |  | 10 | 5.60 |  |
| Undergraduate Studies |  | 105 | 58.70 |  |
| Graduate Studies |  | 34 | 19.00 |  |
| **Total GWG[*]** |  |  |  |  |
| Insufficient |  | 42 | 23.50 |  |
| Adequate |  | 74 | 41.30 |  |
| Excessive |  | 63 | 35.20 |  |
| **Trimester** |  |  |  |  |
| First |  | 30 | 16.80 |  |
| Second |  | 27 | 15.10 |  |
| Third |  | 122 | 68.20 |  |
| **Nutritional Status** |  |  |  |  |
| Very bad |  | 4 | 2.20 |  |
| Bad |  | 9 | 5.00 |  |
| Average |  | 54 | 30.20 |  |
| Good |  | 94 | 52.50 |  |
| Excellent |  | 18 | 10.10 |  |
| **Physical well being** |  |  |  |  |
| Very bad |  | 8 | 4.50 |  |
| Bad |  | 20 | 11.20 |  |
| Average |  | 52 | 29.10 |  |
| Good |  | 81 | 45.30 |  |
| Excellent |  | 18 | 10.10 |  |

[‡] BMI: Body Mass Index

[*]GWG: Gestational weight gain categorized according to Institute of Medicine (IOM) classification [26]

1.7% of women participated in high intensity/vigorous exercise, 51.4% conducted light intensity of exercise, while 18% had a sedentary lifestyle.

Significant associations were observed between the type and intensity of PA and the educational level (No higher education/ higher education) of our participants. Hence, a p value less than 0.01 was seen in women who engaged in sedentary type of exercise and having a university or a higher degree, while a p value of 0.038 was found for those who attended only school and were more active for household chores. When all the types of activities were summed up, women who had achieved a university degree were physically more active, than non-educated ones (p value 0.037). Summary of those results are presented in Table 2.

The next table (Table 3) indicates the correlations between the type and the intensity of physical activity depending on the gestational trimesters. Women in their last trimester of

**Table 2. Type and intensity of PA, deducted from the PPAQ, in women categorized depending on their educational level (N = 179).**

| | No higher education (n = 40) | Higher education (n = 139) | p value[*] |
|---|---|---|---|
| **Total score of PPAQ (MET.h/wk)**[**] | 181.21 ± 89.73 | 210.89 ± 85.30 | 0.037[*] |
| **By intensity** | | | |
| **Sedentary** | 62.70 ± 42.64 | 95.80 ± 43.78 | <0.01[*] |
| **Light** | 89.91 ± 45.72 | 82.48 ± 45.71 | 0.366 |
| **Moderate** | 28.40 ± 52.46 | 32.44 ± 45.69 | 0.635 |
| **Vigorous** | 0.21 ± 0.84 | 0.18 ± 0.81 | 0.583 |
| **By type** | | | |
| **Household/Caregiving** | 93.29 ± 59.47 | 72.68 ± 53.54 | 0.038[*] |
| **Occupational** | 29.58 ± 54.52 | 65.90 ± 58.93 | 0.001[*] |
| **Sports/exercise** | 2.38 ± 5.71 | 4.58 ± 9.92 | 0.036[*] |

[*] Statistical analysis by ANOVA / Kruskal-Wallis tests with a p value less than 0.05 considered as significant

[**]MET metabolic equivalent turnover

gestation had a more sedentary lifestyle (p value 0.001) and were oriented mostly towards occupational activities (p value 0.001), compared with those in their first or second trimester of pregnancy. In addition, household chores were significantly higher in women, in their second trimester (p value 0.031).

The next two tables (Tables 4 and 5) show the correlations between the personal perception of nutritional status and physical well-being in relation to the type and the intensity of PA performed by participating candidates. As explained previously, women were asked to rate subjectively their nutritional and physical status; those who spent more energy for occupational purposes, had classified themselves as having a poor nutritional status (p value 0.021). Even though only 7% of participants felt having a bad nutritional status, but when correlated to the total PA levels, statistically significant associations were found (p value 0.019). Sports/ exercise activities were significantly performed by women that considered themselves as having a good and an excellent physical status (p value 0.03).

The next table (Table 6) exposes the correlations between age, crowding index, pregestational BMI and the type and intensity of PA. Significant correlations were observed regarding maternal age and occupational activities (p value 0.004) and crowding index and household

**Table 3. Type and intensity of PA, deducted from the PPAQ, in women categorized depending on their gestational trimesters (N = 179).**

| | First trimester (n = 30) | Second trimester (n = 27) | Third trimester (n = 122) | p value[*] |
|---|---|---|---|---|
| **Total score of PPAQ (MET.h/wk)**[**] | 190.65 ± 96.70 | 191.27 ± 76.10 | 210.48 ± 86.67 | 0.377 |
| **By intensity** | | | | |
| **Sedentary** | 75.33 ± 50.70 | 65.46 ± 36.35 | 96.70 ± 43.86 | 0.001[*] |
| **Light** | 86.83 ± 36.02 | 96.24 ± 40.11 | 80.80 ± 48.66 | 0.267 |
| **Moderate** | 28.04 ± 39.67 | 29.39 ± 43.17 | 32.87 ± 50.07 | 0.855 |
| **Vigorous** | 0.45 ± 1.30 | 0.18 ± 0.94 | 0.12 ± 0.61 | 0.151 |
| **By type** | | | | |
| **Household/Caregiving** | 87.77 ± 56.35 | 97.99 ± 42.07 | 70.13 ± 56.59 | 0.031[*] |
| **Occupational** | 37.78 ± 49.81 | 29.01 ± 51.87 | 69.07 ± 60.59 | 0.001[*] |
| **Sports/exercise** | 3.45 ± 5.13 | 2.46 ± 2.98 | 4.60 ± 10.72 | 0.503 |

[*]Statistical analysis by ANOVA / Kruskal-Wallis tests with a p value less than 0.05 considered as significant

[**]MET metabolic equivalent turnover

**Table 4. Type and intensity of PA, deducted from the PPAQ, in women categorized depending on their subjective perception of nutritional status (N = 179).**

|  | Very bad/bad nutritional status (n = 13) | Average nutritional status (n = 54) | Good/Excellent nutritional status (n = 112) | p value[*] |
|---|---|---|---|---|
| **Total score of PPAQ (MET.h/wk)**[**] | 261.95 ± 116.51 | 187.94 ± 63.53 | 205.44 ± 90.59 | 0.021[*] |
| **By intensity** |  |  |  |  |
| Sedentary | 110.53 ± 42.41 | 83.36 ± 45.31 | 88.27 ± 45.65 | 0.155 |
| Light | 103.56 ± 56.69 | 80.50 ± 36.10 | 83.64 ± 48.23 | 0.260 |
| Moderate | 47.48 ± 74.52 | 23.89 ± 38.74 | 33.37 ± 46.78 | 0.216 |
| Vigorous | 0.38 ± 1.35 | 0.18 ± 0.76 | 0.16 ± 0.77 | 0.681 |
| **By type** |  |  |  |  |
| Household/Caregiving | 88.35 ± 30.96 | 78.44 ± 47.02 | 75.45 ± 58.74 | 0.720 |
| Occupational | 97.99 ± 85.69 | 46.55 ± 50.98 | 58.53 ± 58.69 | 0.019[*] |
| Sports/exercise | 4.52±5.86 | 2.78 ± 5.54 | 4.67 ± 10.75 | 0.459 |

[*] Statistical analysis by ANOVA / Kruskal-Wallis tests with a p value less than 0.05 considered as significant

[**]MET metabolic equivalent turnover

chores (p value 0.008). However, no significant relationship was seen between the type and the intensity of exercise and pregestational BMI.

Following a multiple regression analysis, neither maternal age nor the educational level were associated with the total score of PPAQ among the participants. Self-assessed nutritional and physical status were declared significantly correlated with the PA levels (p values 0.007 and 0.022 significantly). All results are presented in Table 7.

## Discussion

This study aimed in translating to Arabic, a tool, used in various researches across Europe, Asia and North American countries and to cross-culturally adapt it to women living in Northern Africa, the Gulf region and the Middle-East. Two certified translators were given the task to translate and back-translate the English version of PPAQ into Arabic. The advantages of this newly translated PPAQ are multiple; it's a simple, rapid and self-administered tool that takes into account the contribution of different daily activities often neglected in other PA

**Table 5. Type and intensity of PA, deducted from the PPAQ, in women categorized depending on their subjective perception of physical well-being (N = 179).**

|  | Very bad/bad physical state (n = 28) | Average physical state (n = 52) | Good/Excellent physical state (n = 99) | p value[*] |
|---|---|---|---|---|
| **Total score of PPAQ (MET.h/wk)**[**] | 176.37 ± 67.44 | 202.37 ± 78.58 | 213.14 ± 94.65 | 0.140 |
| **By intensity** |  |  |  |  |
| Sedentary | 78.78 ± 35.00 | 89.26 ± 42.65 | 90.68 ± 49.53 | 0.471 |
| Light | 73.26 ± 40.92 | 87.13 ± 46.08 | 85.64 ± 46.73 | 0.386 |
| Moderate | 24.33 ± 29.54 | 25.79 ± 40.82 | 36.59 ± 53.61 | 0.279 |
| Vigorous | 0.00 ± 0.00 | 0.19 ± 0.77 | 0.23 ± 0.95 | 0.408 |
| **By type** |  |  |  |  |
| Household/Caregiving | 81.96 ± 54.07 | 70.23 ± 48.63 | 79.67 ± 59.20 | 0.544 |
| Occupational | 39.49 ± 45.86 | 64.83 ± 65.45 | 59.54 ± 59.55 | 0.155 |
| Sports/exercise | 1.48 ± 5.14 | 2.43 ± 3.50 | 5.69 ± 11.55 | 0.03[*] |

[*]Statistical analysis by ANOVA / Kruskal-Wallis tests with a p value less than 0.05 considered as significant

[**]MET metabolic equivalent turnover

**Table 6. Correlations between the type and the intensity of physical activity, deducted from the PPAQ, in women with maternal age, crowding index and pre-gestational BMI (N = 179).**

| | Maternal age | | Crowding Index | | Pre-gestational BMI | |
|---|---|---|---|---|---|---|
| | R | p value* | r | p value* | r | p value* |
| Total score of PPAQ (MET.h/wk)** | 0.113 | 0.132 | 0.093 | 0.217 | -0.007 | 0.930 |
| **By intensity** | | | | | | |
| Sedentary | -0.024 | 0.748 | -0.073 | 0.330 | -0.069 | 0.356 |
| Light | 0.085 | 0.256 | 0.133 | 0.076 | 0.050 | 0.505 |
| Moderate | 0.101 | 0.178 | 0.052 | 0.491 | -0.031 | 0.678 |
| Vigorous | -0.136 | 0.069 | -0.059 | 0.436 | 0.067 | 0.375 |
| **By type** | | | | | | |
| Household/Caregiving | 0.035 | 0.643 | 0.197 | 0.008* | 0.078 | 0.300 |
| Occupational | 0.214 | 0.004* | -0.027 | 0.725 | -0.094 | 0.212 |
| Sports/exercise | -0.139 | 0.063 | -0.106 | 0.156 | -0.128 | 0.087 |

*Statistical analysis by Spearman and Pearson correlation coefficient with a p value less than 0.05 considered as significant

**MET metabolic equivalent turnover

questionnaires [28]. Plus, the sum of its components determines MET values, easily representing "true" activity levels.

Following this part, the research team deducted from the final version, two questions not compatible with the societal and living lifestyle of people in this area of the world, such as mowing the lawn and showering the snow off. Other improvements concerning the lay-out of the tool were adopted too. After a pretest, the newly translated version was tested to evaluate the PA status of 179 pregnant women, recruited from different educational and socio-economic background.

Our sample size, age, gestational trimesters, and pre-gestational BMI were similar to the study of Xiang Mi et al., which used the Chinese version of PPAQ, among 182 pregnant women, having mostly an initial BMI in the normal category [29].

Patterns of PA evolve during pregnancy. A review published by Forczek et al. in 2017 observed a net decline in activities in the first trimester of pregnancy, followed by a rise, after the 14th week of gestation, with women mostly oriented towards occupational activities, housework/caregiving, active lifestyle combining essentially aerobic sports, such as walking and stationary cycling [30]. On the other hand, in a sample of Polish pregnant women, no differences were observed in terms of total PA, across the trimesters of pregnancy [7]. In our sample, the intensity and the type of PA increased in women of second and third trimester, compared to those in their first trimester. Our results join those published by Santos et al. [31], where

**Table 7. Multiple regression analysis of explanatory factors associated with the total score of PPAQ in women.**

| | Unstandardized Coefficients | | Standardized Coefficients | t | Sig. | 95.0% Confidence Interval for B | |
|---|---|---|---|---|---|---|---|
| | B | Std. Error | Beta | | | Lower Bound | Upper Bound |
| (Constant) | 176.566 | 71.302 | | 2.476 | 0.014 | 35.831 | 317.300 |
| Maternal age | 1.654 | 1.281 | .096 | 1.291 | 0.198 | -.875 | 4.183 |
| Education | 20.560 | 15.499 | .099 | 1.327 | 0.186 | -10.031 | 51.150 |
| Nutritional status [Ref: very bad/bad] | -70.038 | 25.462 | -.202 | -2.751 | 0.007* | -120.294 | -19.782 |
| Physical status [Ref: very bad / Bad] | 41.538 | 17.988 | .171 | 2.309 | 0.022* | 6.034 | 77.043 |

*p value less than 0.05 considered as significant

women in their third trimester reported mostly a tendency towards sedentary lifestyle, dominated by occupational type of PA, whereas those in their second trimester had a light level of PA and were oriented more towards household activities. Indeed, the second trimester of pregnancy is considered to be the most comfortable period during gestation, where women often feel more energetic and eager to be physically active [7]. Although, PA may decline before delivery due to excess body weight, lower back pain and pressure caused by the expansion of the uterus [32], our study demonstrated an elevated intensity of PA during the last trimester among ladies for occupational oriented activities. This highlights that working moms stay active before gestation, outside and inside their homes, to accomplish all professional and family duties, before delivery. It should be noted that laws in the Middle East allow only a relatively short period of maternity leave, to be taken after birth. In this research, vigorous levels of PA, mostly conducted in sports or exercise context, were not observed across the trimesters of pregnancy. These results join those obtained in the Chinese study, where few pregnant women reported engaging in high intensity/vigorous activities [29]. Similar results were obtained by Yi-Li Ko et al. in a cohort of 150 pregnant women in Taiwan [33].

Another issue is the difficulty to assess low intensity PA, since people often neglect daily routine chores (such as tidying the house, light cleaning, taking care of a child or an animal), because such activities become habits over time and will no more be considered or felt by the respondent as a way of exercising [34]. Thus, the added value of the PPAQ is the inclusion of normal daily low intensity tasks (household and childcare), frequently assumed in Middle-Eastern countries by women, as part of their responsibilities toward their family. Light activities were prevalent among 51.4% of our participants', as mentioned in a similar cross-cultural adaptation of PPAQ, among Brazilian women, with comparable results [35]. The reason may be that those women initially had a sedentary lifestyle or due to cultural pressure and believes were afraid to start a rigorous PA.

Being more educated and having a university degree were among the maternal factors triggering women to adopt a healthier lifestyle and engage in sport activities. Hence, in our sample, educated women were often more sedentary during work time, but engaged more in sport activities during their free time, compared to less educated housewives (p value 0.036). In our research, the age of the participants was another factor influencing patterns of PA. As women get older, occupational activities tend to increase, most probably due to an increase in their professional responsibilities. Besides, household activities were mostly observed in women having larger families, with a higher crowding index. The reason behind is the numerous home duties of Middle-Eastern females and their responsibilities towards their larger family. However, following a multiple regression analysis, those two variables (educational level and age) were no more significantly correlated with the total PPAQ score, unlike the results of Nascimento et al. in Brazil, where educational level was among the maternal factors inducing an active lifestyle [32]. On the other hand, after controlling cofounding variables, self-rated physical and nutritional well-being remained significantly associated with PA. Future areas of research should focus on the evaluation of the quality of life and the nutritional status by administering validated questionnaires among pregnant women of Arabic descent.

Our findings showed a net decline in sports/exercise related activities irrespective of gestational age or BMI. Watching television was the sedentary behavior that the majority of women spent doing, similar to the study of Santos et al [31] and Chasan-Taber et al [16]. Chandonnet et al. applied the French version of PPAQ among pregnant obese women and showed that they were predominantly active in household and caregiving activities [36]. Although our sample was mostly composed of women with a normal BMI, housewives or those with no higher academic education were engaged mostly in household and caregiving activities.

In parallel, our translated tool (PPAQ-Arabic) was used among another sample of 141 Lebanese pregnant women in another research aiming to study the relationship between their quality of life, insomnia, depression and PA. Published results of that study showed positive correlations between maternal age and total PA (p value 0.034), with no associations seen between PPAQ scores, pregestational BMI, gestational age, worry and insomnia [37].

Our study has some limitations. First, there are possible sampling and recall bias, because the randomly selected participants filled the PPAQ at one single moment of their gestation and PA may change, evolve or decrease during the entire pregnancy. Secondly, MET values used in the calculations were not specific for pregnancy, since Compendium-based MET values were used to estimate the intensity of PPAQ activities. However, this limitation is to be neglected, since those conversion factors were recommended by the original creator of the PPAQ and all other authors who translated this instrument in their native language, applied it too. In addition, the PPAQ is a subjective technique of assessing the PA levels during pregnancy. Better results can be obtained if coupled with a pedometer or an accelerometer that offer a more objective assessment. The future aim of our team is to assess the validity of our tool against accelerometers in a sample of pregnant women. Our sampling approach was not representative of all Lebanese women since we followed a convenient sampling strategy, however it included participants coming from different Lebanese regions, irrespective of their educational, socioeconomic and religious backgrounds. The strengths of our study are the participants being distributed among various gestational weeks and having a normal BMI. In addition, the compliance of all the participants and the easiness to fill this newly adapted tool, with no missing data or dropouts were an added asset of this research. Moreover, seasonal variation did not influence results reflecting to sports / exercise questions, since Lebanon is known for its mild Mediterranean climate, proper to the practice of different type of sports, all year long.

## Conclusion

The ACOG recommends pregnant women with low risk pregnancies to maintain a moderate-intensity level of exercise for the prevention of excessive gestational weight gain [38], optimal fetal outcomes, and maternal well-being [4]. Nevertheless, assessment of PA is hard to interpret and presents many limitations, since there is no gold standard for a specific and a validated tool to be administered exclusively during pregnancy [39]. Following this study, the translation to the Arabic language and the cultural adaptation of the original PPAQ will help researchers depict the PA levels of pregnant women in twenty two Arabic speaking countries, across Asia and Africa.

Because pregnancy is full of physiological and psychological changes, a decline in PA levels is not surprising. Earlier counseling during prenatal services on the advantages of PA is crucial, since this period represents a unique event in women's life and an ideal moment to implement strategies and guidelines to promote and install new health focused behaviors, such as exercise.

## Supporting information

**S1 File. Pregnancy Physical Activity Questionnaire (PPAQ)–Arabic version.**
(DOCX)

## Acknowledgments

The authors gratefully acknowledge the participants and the gynecologists.

## Author Contributions

**Conceptualization:** Tatiana Papazian, Lydia Rabbaa Khabbaz.

**Data curation:** Nada El Osta, Hala Hout, Daisy El Chammas.

**Formal analysis:** Nada El Osta.

**Investigation:** Tatiana Papazian, Hala Hout.

**Methodology:** Tatiana Papazian, Nour El Helou.

**Project administration:** Lydia Rabbaa Khabbaz.

**Resources:** Georges Abi Tayeh, Lydia Rabbaa Khabbaz.

**Software:** Nada El Osta.

**Supervision:** Tatiana Papazian, Hassan Younes, Georges Abi Tayeh, Lydia Rabbaa Khabbaz.

**Validation:** Tatiana Papazian, Hassan Younes, Georges Abi Tayeh, Lydia Rabbaa Khabbaz.

**Visualization:** Tatiana Papazian.

**Writing – original draft:** Tatiana Papazian.

**Writing – review & editing:** Lydia Rabbaa Khabbaz.

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
