## [Decision Letter · Decision Letter 0]

11 Nov 2019

PONE-D-19-25377

Pregnancy physical activity questionnaire (PPAQ): translation and cross cultural adaption of an Arabic version.

PLOS ONE

Dear Dr Papazian,

Thank you for submitting your manuscript to PLOS ONE. After careful consideration, we feel that it has merit but does not fully meet PLOS ONE’s publication criteria as it currently stands. Therefore, we invite you to submit a revised version of the manuscript that addresses the points raised during the review process.

General comments

The manuscript is not presented in an intelligible fashion and typographical errors are commonly encountered.The approach used for classifying some of your key variables (e.g. gestational weight gain, nutritional status and physical well being) has not been clearly described at all.Please make sure that you have followed the PLOS one standards in writing the in-text citations and references.Please revise the discussion section as per the recommendation provided by the two reviewers. 

We would appreciate receiving your revised manuscript by Dec 26 2019 11:59PM. To enhance the reproducibility of your results, we recommend that if applicable you deposit your laboratory protocols in protocols.io, where a protocol can be assigned its own identifier (DOI) such that it can be cited independently in the future. For instructions see: http://journals.plos.org/plosone/s/submission-guidelines#loc-laboratory-protocols

We look forward to receiving your revised manuscript.

Kind regards,

Samson Gebremedhin, PhD

Academic Editor

PLOS ONE

Journal Eequirements:

2. Please include additional information regarding the sample size of your main study and how this number of participants was reached - e.g. referring to previous studies or a power calculation.

This research received no specific grant from any funding agency, commercial or not-for-profit sectors.

5. Please amend the manuscript submission data (via Edit Submission) to include author Daisy Chammas.

6. We note you have included a table to which you do not refer in the text of your manuscript. Please ensure that you refer to Table 6 in your text; if accepted, production will need this reference to link the reader to the Table.

Additional Editor Comments (section-by-section comments):

Abstract

Please make sure that the abstract is prepared in line with the standards of the journal.Can you please provide additional information on how the “translation, pre-testing, and cross-cultural adaptation” was made?It would be good to change the header “discussion” to “conclusion”

Background

Line 44-47: the sentence “The Pregnancy Physical Activity Questionnaire (PPAQ) is a self-administered tool, created to evaluate the frequency, duration and intensity of different types of activities conducted by pregnant females during the last three months.” Is vague? When you say last 3 months, do you mean the last trimester or in the preceding 3 months of the survey?

Methods

Did you evaluate the adequacy of the sample size for measuring the level of physical activity in the study population? In the manuscript nothing has been said about it.Line 121: the PPAQ items were dropped to 31. Do you think this had implication on the level of physical activity you estimated in your population? As you reduce the item have you proportionally reduced the cut-off for defining the level of physical activity?Page 130-31: how did you assure that pregnant females from different educational backgrounds, socioeconomic status and gestational age were recruited? Did you apply kind of purposive sampling? And how did this affect the external validity of your study?Line 132-33: did you observe considerable difference between the information collected by interviewer-administered and self-administered questionnaires? Nothing has been said about this in the results and discussion sections.

Results

Table 7: please report the means and the standard deviations with the same precision.Table 7, do you really need 5 decimal place number for the crowding index.In the methods section please describe how you computed the variable “crowding index”.Table 7: seems few of the study subjects were illiterates. So how did they fill the self-administered version of the questionnaire? I think they should have been excluded from the study in the first place.Table 7: How did you classify gestational weight gain as insufficient, adequate or excessive? Similarly, the approach used to classify the variables “nutritional status” and “physical wellbeing” is confusing because no operational definitions had been given.Line 170: I think you need to have another sub-header here. Information about MET should not be under socio-demographic information176-78: “Significant correlations were observed when we stratified the sample depending on …….” Correlation between what?Line 178: “a p value of 0.00” >> “p value < 0.01”Table 3: please clearly indicate with * all the significant p values.Line 221: “The last table ” >> “Table 6”Table 6: what was the basis for selecting the three variables (maternal age, crowding index and pregestational BMI) as key predictors of physical activity during pregnancy? Have you had a priority hypothesis or was that a post-hoc analysis?Again, it is not clear how the pre-pregnancy BMI was determined. How did you manage to get information about their pre-pregnancy weight?

Discussion

Please see blow the comments forwarded by the reviewers

Conclusion

Line 301-5: I think this should rather be in the discussion, not conclusion, section

Reference and citation

Please note that PLOS uses “Vancouver” style, as outlined here https://www.nlm.nih.gov/bsd/uniform_requirements.html: please modify the intext citations and references accordingly.

Reviewers' comments:

Reviewer's Responses to Questions

**Comments to the Author**

1. Is the manuscript technically sound, and do the data support the conclusions?

Reviewer #1: No

Reviewer #2: Partly

2. Has the statistical analysis been performed appropriately and rigorously? 

Reviewer #1: Yes

Reviewer #2: No

3. Have the authors made all data underlying the findings in their manuscript fully available?

Reviewer #1: No

Reviewer #2: Yes

4. Is the manuscript presented in an intelligible fashion and written in standard English?

Reviewer #1: No

Reviewer #2: No

5. Review Comments to the Author

Reviewer #1: The authors present an article about an interesting topic: adaptation of The Pregnancy Physical Activity Questionnaire (PPAQ) to Arabian conditions. However, it is necessary to justify the choice of instrument. Maybe it would be more informative the use of a specific questionnaire for pregnant women than a general questionnaire, e.g. Pregnancy Sympton Inventory – PSI (Foxcroft KF, Callaway LK, Byrne NM, Webster J. Development and validation of a pregnancy symptoms inventory. BMC Pregnancy Childbirth. 2013;13:3. Published 2013 Jan 16. doi:10.1186/1471-2393-13-3). The PSI provides a comprehensive inventory of pregnancy related symptoms, with a mechanism for assessing their effect on function. It was robustly developed, with good test re-test reliability, face validity, comprehension and readability. This provides a validated tool for assessing the impact of interventions in pregnancy.

The major mistypes have been detected:

1. The structure of the Abstract does not comply with PLoS One guidelines. The aim is not clearly defined. In addition to adapting PPAQ to Arabic conditions, the purpose of the work is - as I understand it - to study the relationship of physical activity (PA) of pregnant women with their age, crowding index and pregestational body mass index. Abbreviations should not be used in the Abstract. The authors write that they analyze the socioeconomic status - which is not consistent with the paper. By the way, this is a big limitation of this study. Physical activity in pregnancy may depend on the socio-economic status and support of a partner, friends or family.

2. The Introduction does not present the background of the paper. There is no information on the state of knowledge regarding PA in pregnant women. There are many publications on this subject, e.g .:

- Krzepota J., Sadowska D., Biernat E. Relationship between physical activity and quality of life in pregnant woman in the second and third trimester. International Journal of Environmental Research and Public Health 2018: 15: 2745.

- Gjestland, K., Bø, K., Owe, K. M., & Eberhard-Gran, M. (2012). Do pregnant women follow exercise guidelines? Prevalence data among 3482 women, and prediction of low-back pain, pelvic girdle pain and depression. Br J Sports Med, bjsports-2012.

- Fell, D. B., Joseph, K. S., Armson, B. A., & Dodds, L. (2009). The impact of pregnancy on physical activity level. Maternal and Child Health Journal, 13(5), 597–603.

- Evenson, K. R., Savitz, A., & Huston, S. L. (2004). Leisure‐time physical activity among pregnant women in the US. Paediatric and Perinatal Epidemiology, 18(6), 400–407.

Line 48-60 - this fragment should be moved to the Method chapter

3. Statistical Analysis

Line 153 – it is necessary to make the following sentence more specific: 1 MET is the metabolic equivalent of the energy expended at rest and equals 3.5 ml O2/kg/min.

4. Results

Line 184 - missing ”p”, as in line 222

Line 190 –it is necessary to mark significant differences* in tab. 3

Line 199 - full stop is not placed at the end of legend

In Tab. 2-6 it is necessary to standardize the p value record in the legend. In tab. 2 and 3 there is: p value less than 0.05 considered as significant. In tab. 4, 5, 6 - * P < 0.05

Line 205 – standardize the lower case p (e.g. line 182), the capital P (e.g. line 205)

Line 224-226 – this fragment should be moved to Discussion

5. Apart from the Introduction, the Discussion is the weakest part of the paper. The Authors refer to data from 35 years ago (line 236).

Line 237-240 – The sentence Although the ACOG recommends pregnant females to maintain a moderately steady intensity level of exercise, yet a low percentage of pregnant women exercise at recommended levels, which is significantly lower when compared to the general population (“ACOG Committee Opinion No. 650,” 2015) is not clear. Which women? Lebanese?

Line 249 – where does this conclusion come from? As expected, PA increased during the second trimester in our sample. The results indicate that it only refers to household/caregiving and light intensity PA.

Line 258-260 – instead of 52% it should be 51,4%

Line 266 - Watching television was the sedentary behavior that the majority of women spent, similar in the study of Santos PC et al (Santos et al., 2016) and Chasan-Taber et al (Chasan-Taber et al., 2004a). How do we know it? How does it relate to the phrase in line 174? -

18% had a sedentary lifestyle

Line 267 i 268 - the first letter of the name is not necessary

Line 268 – The following sentence raises controversy: Chandonnet N. et al. applied the French version of PPAQ among pregnant obese women who were predominantly active in household and caregiving activities, as in our sample specially among housewives, or those with no higher academic education (Chandonnet et al., 2012). The quoted study can only be compared to obese participants. Have the authors examined the professional status? Where does this conclusion come from? Besides, the quote (Chandonnet et al., 2012) should appear after Chandonnet N. et al. – and not at the end of the sentence, because this part of the sentence regards the study of the Authors.

Line 273-276 –In the sentence Results derived from this study showed positive correlations regarding maternal age and total PA ( P= 0.034), however no associations were seen between PPAQ scores and pregestational BMI, gestational age, worry or insomnia (Mourady et al., 2017) it is not clear which results refer to the study by Mourady et al., 2017, and which refer to the Authors’ study. Moreover, worry and insomnia were not tested.

Reviewer #2: The authors present a work of interest and the translation of a widely used tool during pregnancy, that is useful to use in other countries and languages. Also, it gives a general insight of the PA levels of Lebanese pregnant women for the first time, being a topic of interest. However, the presentation of the results and discussion needs review, as well as some part of the statistical analysis. The full text need a deep revision of the English writing, as well as the construction of several sentences.

Abstract

Line 5. Please change PA level for “levels” throughout the text, as it refers to different intensities.

Line 7-9. Please rephrase the sentence or watch the punctuation, as it is difficult to follow.

Line 13. Please change females for women throughout the text.

Line 19-20. Compare to what? To first trimester? Or to other types of activities? Please specify in abstract. Also, the second trimester does not have the highest results for occupational activities, if you see the respective table.

Introducition

Page 9 line 41. Change benefice for benefits.

Line 41-44. Could you cite any of those questionnaires that are too long or subjective?. How is PPAQ not subjective, as it is a self-administred questionnaire. Please rewrite if necessary.

Line 51. Add to the sentence that is the amount of time spent in the specific activity.

Line 51. Start the sentence with Six instead of 6.

Methods

Line 100. Don’t know if I understand well. The 45 pregnant females were all part of the university staff? If not, please, rephrase.

Line 109. Mentioned where? In the introduction you say PA is important together with nutrition, but no mentioning of using another questionnaire. Furthermore, even if this particular study does not need that specific information as you used another complimentary questionnaire to assess that, maybe another study will use the full information, and therefore, the final Arabic version should take into account all the questions.

Line 118. There are no other activities more usual in the Middle East that could be changed for those activities with a similar MET expenditure? This way the totals could be comparable with studies from other countries using the PPAQ.

Line 120. Please change “our” environment for “Middle Eastern” or “Arabic” countries.

Line 127. Please cite the sutyd of there is a clinical trial or pprevious work published.

Line 129. Singleton baby or singleton pregnancy?

Line 134. Please refer to self-administer the newly translated PPAQ

Line 154. Please specify that the final number was 2 less in your study and you took that into account in the corresponding kind of activity/intensity totals.

Statistical Analyses. Please indicate the corresponding table of results to each of the statistical analyses described.

The authors don’t specify any confounders for the ANOVA analyses comparing the PPAQ results by groups. However, when comparing, for example the PA levels by high education of low education (table 2) they don’t take into account how many women there are of each trimester of gestation, in each group. Even more, given the information that the PA levels differ significantly during the different trimesters, they may consider it as confounder. The same would apply for other tables and confounders.

Results

162 add “of pregnancy” at the end of the sentence.

Line 204-205. Please keep the hypothesis and subjective approach for the discussion section with the proper citation.

Tables. In all titles of the tables, please add “sedentary time” to the “Type and intensity of PA”. Also, please add units of measurement.

Line 222. Add “respectively” after the P values.

Lines 224-228. Please put this information into the discussion section.

Discussion.

Line 251. The authors state “PA increased during the second trimester in our sample. Indeed, this period is considered to be the most comfortable period during gestation, where women often feel themselves more energetic and eager to be physically active”, while in the table, the total score for PPAQ is higher in the third trimester than the second, as well as in some particular intensities and activities. Also, please use references when explaining your results.

Line 253-256. Please rephrase into shorter sentences (there is twice the word “since”)

277. Highlights.

286-287. Final part of the sentence is not clear.

294. How is the sample homogenous if there are women of all gestational ages and from different studies and economic backgrounds?

Conclusions.

Line 303. Please rephrase the last sentence.

Line 310. Women

6. PLOS authors have the option to publish the peer review history of their article (what does this mean?). If published, this will include your full peer review and any attached files.

Reviewer #1: No

Reviewer #2: No

---

## [Author Response · Author response to Decision Letter 0]

25 Dec 2019

PONE-D-19-25377

Pregnancy physical activity questionnaire (PPAQ): translation and cross cultural adaption of an Arabic version.

PLOS ONE

Dear Dr Papazian,

Thank you for submitting your manuscript to PLOS ONE. After careful consideration, we feel that it has merit but does not fully meet PLOS ONE’s publication criteria as it currently stands. Therefore, we invite you to submit a revised version of the manuscript that addresses the points raised during the review process.

General comments

• The manuscript is not presented in an intelligible fashion and typographical errors are commonly encountered.

The article has been reviewed by a colleague who is fluent in English to correct all grammatical and typing mistakes.

• The approach used for classifying some of your key variables (e.g. gestational weight gain, nutritional status and physical well being) has not been clearly described at all.

We added the classifications and details of those key variables in the main manuscript. Thank you.

• Please make sure that you have followed the PLOS one standards in writing the in-text citations and references.

All corrections were done.

• Please revise the discussion section as per the recommendation provided by the two reviewers. 

Done. New references were added as recommended by the reviewers.

We would appreciate receiving your revised manuscript by Dec 26 2019 11:59PM. To enhance the reproducibility of your results, we recommend that if applicable you deposit your laboratory protocols in protocols.io, where a protocol can be assigned its own identifier (DOI) such that it can be cited independently in the future. For instructions see: http://journals.plos.org/plosone/s/submission-guidelines#loc-laboratory-protocols

• A rebuttal letter that responds to each point raised by the academic editor and reviewer(s). This letter should be uploaded as separate file and labeled 'Response to Reviewers'.

• A marked-up copy of your manuscript that highlights changes made to the original version. This file should be uploaded as separate file and labeled 'Revised Manuscript with Track Changes'.

• An unmarked version of your revised paper without tracked changes. This file should be uploaded as separate file and labeled 'Manuscript'.

We look forward to receiving your revised manuscript.

Kind regards,

Samson Gebremedhin, PhD

Academic Editor

PLOS ONE

Journal Eequirements:

Done.

2. Please include additional information regarding the sample size of your main study and how this number of participants was reached - e.g. referring to previous studies or a power calculation.

We included in the final analyses, the data of all participants who were recruited (N=179), even though, initially we aimed to include 150 subjects. This value was arbitrarily set, taking into account the sample sizes used in previous similar studies. The main developer of this tool, Lisa Chasan-Taber administered the newly created tool in a sample of 235 women 1, while Santos et al in 118 Portuguese females, in 2016. Çirak Y. et al translated and evaluated the physical activity level of 205 Turkish women, while in a country as overpopulated as China, this tool was translated and tested among 182 pregnant women Mi Xiang. Chandonnet et al. translated and applied this instrument among 56 French women (Plos one).

This research received no specific grant from any funding agency, commercial or not-for-profit sectors.

Done.

Corrections were done. This study received no funding. 

Done. Corrections were done according to the instructions provided.

5. Please amend the manuscript submission data (via Edit Submission) to include author Daisy Chammas.

Done. Thank you.

6. We note you have included a table to which you do not refer in the text of your manuscript. Please ensure that you refer to Table 6 in your text; if accepted, production will need this reference to link the reader to the Table.

Done. Thank you.

Additional Editor Comments (section-by-section comments):

Abstract

• Please make sure that the abstract is prepared in line with the standards of the journal.

Done.

• Can you please provide additional information on how the “translation, pre-testing, and cross-cultural adaptation” was made?

A brief explanation was added in the abstract.

• It would be good to change the header “discussion” to “conclusion”

Done.

Background

• Line 44-47: the sentence “The Pregnancy Physical Activity Questionnaire (PPAQ) is a self-administered tool, created to evaluate the frequency, duration and intensity of different types of activities conducted by pregnant females during the last three months.” Is vague? When you say last 3 months, do you mean the last trimester or in the preceding 3 months of the survey?

The PPAQ measures physical activity level during the preceding 3 months of the survey. This was corrected in the manuscript. Thank you for your comment.

Methods

• Did you evaluate the adequacy of the sample size for measuring the level of physical activity in the study population? In the manuscript nothing has been said about it.

The primary aim of the study was to translate and culturally adapt an internationally validated tool (PPAQ) into Arabic and the secondary aim was to test it on a sample of pregnant women in a descriptive cross-cultural study. We aimed to include 150 subjects. This value was arbitrarily set, taking into account the sample sizes used in previous similar studies.(METTRE REF)

Our research team aims to validate our tool (PPAQ-arabic) against accelerometers in an upcoming study among a sample size with a more statistical power. 

• Line 121: the PPAQ items were dropped to 31. Do you think this had implication on the level of physical activity you estimated in your population? As you reduce the item have you proportionally reduced the cut-off for defining the level of physical activity?

The removal of those two questions will not affect the total calculations/ cut-off determination, since those items were referring to mowing the lawn and showering the snow off. Those activities are not possible to be realized in the hot, humid, sandy and desert climate of the Middle-East. In addition, since the region is overpopulated, the housing system is mostly in buildings and not private houses, with no gardening facilities. Hence, our research committee presumed that the response “Never” of our participants to those 2 items, will not affect the outcome of our data analyses. 

• Page 130-31: how did you assure that pregnant females from different educational backgrounds, socioeconomic status and gestational age were recruited? Did you apply kind of purposive sampling? And how did this affect the external validity of your study?

We did not apply a kind of purposive sampling. Women were recruited during their prenatal visits, in different clinics during their regular checkups to their gynecologist. After insuring that the patient fulfills the selection criterias, the research team explained to each participant the aim of this research, irrespective of her educational, socioeconomical or gestational age. We aimed to recruit pregnant women at different trimesters, in order to study the evolution of physical activity during gestation. 

• Line 132-33: did you observe considerable difference between the information collected by interviewer-administered and self-administered questionnaires? Nothing has been said about this in the results and discussion sections.

The creation of the Arabic version of PPAQ to evaluate the physical activity level among our participants was part of our global questionnaire, which focused more on Mediterranean diet adherence and nutritional status of pregnant females. The section reserved to PPAQ was self-administered and not filled during the face to face interview. Research assistants were present in the medical cabinet, but their role was to interfere only if a woman was unable to understand an item present in the questionnaire. 

Results

• Table 7: please report the means and the standard deviations with the same precision.

Done.

• Table 7, do you really need 5 decimal place number for the crowding index.

Two decimals were kept after the decimal point. Thank you for your comment.

• In the methods section please describe how you computed the variable “crowding index”.

Done. An explanation was provided.

• Table 7: seems few of the study subjects were illiterates. So how did they fill the self-administered version of the questionnaire? I think they should have been excluded from the study in the first place.

Those subjects were not illiterates, however they had a primary level of education. Hence, we omitted the word “illiterate” and replace it with “primary”. The rate of illiteracy is very low in our country and the Lebanese societal and cultural habits are quite different from the Arabic neighboring nations, since Lebanese women are more socially empowered, which helps them play an important role in our modern society. 

In addition, according to the latest national statistics in our country (in 2007) concerning current educational level according to age, data show that more than 80% of the Lebanese population aged above 19 years has a university degree, even in low income regions. These percentages reach respectively 90 and 92.6% in Lebanese aged between 25-29 and 35-39 (with no significant differences between males and females). Hence, we can say that women in our sample were not illiterate. It was a vocabulary mistake. Thank you for your remark. 

 (Ref: http://www.cas.gov.lb/images/PDFs/Educational%20status-2007-ar.pdf, page 214) 

• Table 7: How did you classify gestational weight gain as insufficient, adequate or excessive? Similarly, the approach used to classify the variables “nutritional status” and “physical wellbeing” is confusing because no operational definitions had been given.

Gestational weight gain was classified according to the values defined by the Institute of Medicine1. This was added in the manuscript with the corresponding reference.

1 Institute of Medicine (US) and National Research Council (US) Committee to Reexamine IOM Pregnancy Weight Guidelines. Weight Gain During Pregnancy: Reexamining the Guidelines [Internet]. Rasmussen KM, Yaktine AL, editors. Washington (DC): National Academies Press (US); 2009 [cited 2019 Mar 13]. (The National Academies Collection: Reports funded by National Institutes of Health). Available from: http://www.ncbi.nlm.nih.gov/books/NBK32813/

Women rated their “Nutritional status” and “Physical wellbeing” subjectively, by providing answers in a Likert scale; very bad, bad, average, good and excellent. 

• Line 170: I think you need to have another sub-header here. Information about MET should not be under socio-demographic information

Done. We added a sub-header as requested. Thank you.

• 176-78: “Significant correlations were observed when we stratified the sample depending on …….” Correlation between what?

Done. The correction was done according to the remark. Thank you.

• Line 178: “a p value of 0.00” >> “p value < 0.01”

Done.

• Table 3: please clearly indicate with * all the significant p values.

Done.

• Line 221: “The last table ” >> “Table 6”

Done.

• Table 6: what was the basis for selecting the three variables (maternal age, crowding index and pregestational BMI) as key predictors of physical activity during pregnancy? Have you had a priority hypothesis or was that a post-hoc analysis?

The selection of those variables were highlighted and emphasized after finding significant associations with physical activity. We did not have a priority hypothesis, prior to that outcomes.

• Again, it is not clear how the pre-pregnancy BMI was determined. How did you manage to get information about their pre-pregnancy weight?

BMI was determined as the ratio of pregestational weight in kgs to the square of height in meter. It was then categorized according to the WHO cut-off points1. All anthropometric measurements were copied directly from the medical file, the day of the recruitment. Details and references concerning this issue were added in the manuscript. Thank you.

1Physical status: the use and interpretation of anthropometry. Report of a WHO Expert Committee. World Health Organ Tech Rep Ser. 1995;854:1–452.

Discussion

• Please see blow the comments forwarded by the reviewers

Done. 

Conclusion

• Line 301-5: I think this should rather be in the discussion, not conclusion, section

Done. Thank you.

Reference and citation

• Please note that PLOS uses “Vancouver” style, as outlined here https://www.nlm.nih.gov/bsd/uniform_requirements.html: please modify the intext citations and references accordingly.

Done. All references were written in Vancouver style. 

Reviewers' comments:

Reviewer's Responses to Questions

Comments to the Author

1. Is the manuscript technically sound, and do the data support the conclusions?

Reviewer #1: No

Reviewer #2: Partly

2. Has the statistical analysis been performed appropriately and rigorously? 

Reviewer #1: Yes

Reviewer #2: No

3. Have the authors made all data underlying the findings in their manuscript fully available?

Reviewer #1: No

Reviewer #2: Yes

4. Is the manuscript presented in an intelligible fashion and written in standard English?

Reviewer #1: No

Reviewer #2: No

---

## [Editor Report · Decision Letter 1]

14 Jan 2020

PONE-D-19-25377R1

Pregnancy physical activity questionnaire (PPAQ): translation and cross cultural adaption of an Arabic version.

PLOS ONE

Dear Dr Papazian,

Thank you for submitting your manuscript to PLOS ONE. After careful consideration, we feel that it has merit but does not fully meet PLOS ONE’s publication criteria as it currently stands. Therefore, we invite you to submit a revised version of the manuscript that addresses the points raised during the review process.

Abstract

Line 7-9: Please also indicate one of the objectives of the study was to measure the physical activity of pregnant women using the adapted PPAQ.The methods sub-section is very superficial and does not clearly describe the four steps followed in the adaptation process.The results sub-section looks deficient because it does not report on the cross-cultural adaptations made to the PPAQ tool, which is one of the objectives of the study.  Please revise accordingly.“with a significant p-value of 0.031 and 0.001 respectively” which specific groups are these p-values referring to? Household? Caregiving? occupational activities?“Our results reported that 51% of women engage in light intensity physical activity. Please also report % who had high intensity exercise and sedentary life style.

Background

Line 38-9: Can you put a citation for the claim “In addition, due to religious sensitivities, exercising in public areas or local gyms are forbidden for women in some Middle-Eastern countries”.

Methods

Line 76-76: please provide citation for the sentence “Our research team followed the guidelines depicted by Guillemin et al., which were composed of 4 phases, to guarantee the reliability of the translated tool”.In the methods section please clearly describe how the 179 student participants were selected for the study and how this sample size was reached at.

Results

Line 177-8: “The mean BMI calculated based on their pregestational weight was 22.94± 3.60 kg/m” how did you manage to determine the pre-pregnancy weight? Please describe in the methods section.Line 180-182 and table-1: it is not clear what you mean by “self-reported nutritional status” and “self-reported physical well-being”. Please also clearly describe in the methods section how these variables were measured.   Line 192: “Significant correlations……” please change to “significant association”.

Discussion

Line 250-1: “This ensured the homogeneity of our sample.”??? Not clear please rephrase it again?Line 315 “The strength of our study is the homogeneity of the sample”. Homogeneity on the basis of what?

We would appreciate receiving your revised manuscript by Feb 28 2020 11:59PM. To enhance the reproducibility of your results, we recommend that if applicable you deposit your laboratory protocols in protocols.io, where a protocol can be assigned its own identifier (DOI) such that it can be cited independently in the future. For instructions see: http://journals.plos.org/plosone/s/submission-guidelines#loc-laboratory-protocols

We look forward to receiving your revised manuscript.

Kind regards,

Samson Gebremedhin, PhD

Academic Editor

PLOS ONE

Additional Editor Comments (if provided):

Please also make sure that the following comments, which had been forward by one of the reviewers during the first-round review, had been accommodated.

The authors present a work of interest and the translation of a widely used tool during pregnancy, that is useful to use in other countries and languages. Also, it gives a general insight of the PA levels of Lebanese pregnant women for the first time, being a topic of interest. However, the presentation of the results and discussion needs review, as well as some part of the statistical analysis. The full text need a deep revision of the English writing, as well as the construction of several sentences.

Abstract

Line 5. Please change PA level for “levels” throughout the text, as it refers to different intensities.

Line 7-9. Please rephrase the sentence or watch the punctuation, as it is difficult to follow.

Line 13. Please change females for women throughout the text.

Line 19-20. Compare to what? To first trimester? Or to other types of activities? Please specify in abstract. Also, the second trimester does not have the highest results for occupational activities, if you see the respective table.

Introducition

Page 9 line 41. Change benefice for benefits.

Line 41-44. Could you cite any of those questionnaires that are too long or subjective?. How is PPAQ not subjective, as it is a self-administred questionnaire. Please rewrite if necessary.

Line 51. Add to the sentence that is the amount of time spent in the specific activity.

Line 51. Start the sentence with Six instead of 6.

Methods

Line 100. Don’t know if I understand well. The 45 pregnant females were all part of the university staff? If not, please, rephrase.

Line 109. Mentioned where? In the introduction you say PA is important together with nutrition, but no mentioning of using another questionnaire. Furthermore, even if this particular study does not need that specific information as you used another complimentary questionnaire to assess that, maybe another study will use the full information, and therefore, the final Arabic version should take into account all the questions.

Line 118. There are no other activities more usual in the Middle East that could be changed for those activities with a similar MET expenditure? This way the totals could be comparable with studies from other countries using the PPAQ.

Line 120. Please change “our” environment for “Middle Eastern” or “Arabic” countries.

Line 127. Please cite the sutyd of there is a clinical trial or pprevious work published.

Line 129. Singleton baby or singleton pregnancy?

Line 134. Please refer to self-administer the newly translated PPAQ

Line 154. Please specify that the final number was 2 less in your study and you took that into account in the corresponding kind of activity/intensity totals.

Statistical Analyses. Please indicate the corresponding table of results to each of the statistical analyses described.

The authors don’t specify any confounders for the ANOVA analyses comparing the PPAQ results by groups. However, when comparing, for example the PA levels by high education of low education (table 2) they don’t take into account how many women there are of each trimester of gestation, in each group. Even more, given the information that the PA levels differ significantly during the different trimesters, they may consider it as confounder. The same would apply for other tables and confounders.

Results

162 add “of pregnancy” at the end of the sentence.

Line 204-205. Please keep the hypothesis and subjective approach for the discussion section with the proper citation.

Tables. In all titles of the tables, please add “sedentary time” to the “Type and intensity of PA”. Also, please add units of measurement.

Line 222. Add “respectively” after the P values.

Lines 224-228. Please put this information into the discussion section.

Discussion.

Line 251. The authors state “PA increased during the second trimester in our sample. Indeed, this period is considered to be the most comfortable period during gestation, where women often feel themselves more energetic and eager to be physically active”, while in the table, the total score for PPAQ is higher in the third trimester than the second, as well as in some particular intensities and activities. Also, please use references when explaining your results.

Line 253-256. Please rephrase into shorter sentences (there is twice the word “since”)

277. Highlights.

286-287. Final part of the sentence is not clear.

294. How is the sample homogenous if there are women of all gestational ages and from different studies and economic backgrounds?

Conclusions.

Line 303. Please rephrase the last sentence.

Line 310. Women
---

## [Author Response · Author response to Decision Letter 1]

19 Feb 2020

Rebuttal Letter 

Dear editor and reviewers,

On behalf of all the authors, we highly appreciate your valuable comments and recommendations, that were fully taken into consideration in the final revised submitted manuscript. Hope it will meet your requirements. 

PONE-D-19-25377R1

Pregnancy physical activity questionnaire (PPAQ): translation and cross cultural adaption of an Arabic version.

PLOS ONE

Dear Dr Papazian,

Thank you for submitting your manuscript to PLOS ONE. After careful consideration, we feel that it has merit but does not fully meet PLOS ONE’s publication criteria as it currently stands. Therefore, we invite you to submit a revised version of the manuscript that addresses the points raised during the review process.

Abstract

• Line 7-9: Please also indicate one of the objectives of the study was to measure the physical activity of pregnant women using the adapted PPAQ. Done.

• The methods sub-section is very superficial and does not clearly describe the four steps followed in the adaptation process. An explanation was added.

• The results sub-section looks deficient because it does not report on the cross-cultural adaptations made to the PPAQ tool, which is one of the objectives of the study. Please revise accordingly. Done.

• “with a significant p-value of 0.031 and 0.001 respectively” which specific groups are these p-values referring to? Household? Caregiving? occupational activities? An explanation was added.

• “Our results reported that 51% of women engage in light intensity physical activity. Please also report % who had high intensity exercise and sedentary life style. Done.

Background

• Line 38-9: Can you put a citation for the claim “In addition, due to religious sensitivities, exercising in public areas or local gyms are forbidden for women in some Middle-Eastern countries”. A citation was added.

Methods

• Line 76-76: please provide citation for the sentence “Our research team followed the guidelines depicted by Guillemin et al., which were composed of 4 phases, to guarantee the reliability of the translated tool”. Done.

• In the methods section please clearly describe how the 179 student participants were selected for the study and how this sample size was reached at. An explanation was provided. 

Results

• Line 177-8: “The mean BMI calculated based on their pregestational weight was 22.94± 3.60 kg/m” how did you manage to determine the pre-pregnancy weight? Please describe in the methods section. An explanation was provided. 

• Line 180-182 and table-1: it is not clear what you mean by “self-reported nutritional status” and “self-reported physical well-being”. Please also clearly describe in the methods section how these variables were measured. An explanation was provided. 

• Line 192: “Significant correlations……” please change to “significant association”. Done

Discussion

• Line 250-1: “This ensured the homogeneity of our sample.”??? Not clear please rephrase it again?

• Line 315 “The strength of our study is the homogeneity of the sample”. Homogeneity on the basis of what?

Corrections were done and the term “homogeneity” was removed from the manuscript. Thank you.

We would appreciate receiving your revised manuscript by Feb 28 2020 11:59PM. To enhance the reproducibility of your results, we recommend that if applicable you deposit your laboratory protocols in protocols.io, where a protocol can be assigned its own identifier (DOI) such that it can be cited independently in the future. For instructions see: http://journals.plos.org/plosone/s/submission-guidelines#loc-laboratory-protocols

• A rebuttal letter that responds to each point raised by the academic editor and reviewer(s). This letter should be uploaded as separate file and labeled 'Response to Reviewers'.

• A marked-up copy of your manuscript that highlights changes made to the original version. This file should be uploaded as separate file and labeled 'Revised Manuscript with Track Changes'.

• An unmarked version of your revised paper without tracked changes. This file should be uploaded as separate file and labeled 'Manuscript'.

We look forward to receiving your revised manuscript.

Kind regards,

Samson Gebremedhin, PhD

Academic Editor

PLOS ONE

Additional Editor Comments (if provided):

Please also make sure that the following comments, which had been forward by one of the reviewers during the first-round review, had been accommodated.

The authors present a work of interest and the translation of a widely used tool during pregnancy, that is useful to use in other countries and languages. Also, it gives a general insight of the PA levels of Lebanese pregnant women for the first time, being a topic of interest. However, the presentation of the results and discussion needs review, as well as some part of the statistical analysis. The full text need a deep revision of the English writing, as well as the construction of several sentences.

The manuscript was corrected by a colleague, whose native language was English. A regression analysis was done to improve the statistical outcomes and the results. The discussion section was reviewed as well. 

Abstract

Line 5. Please change PA level for “levels” throughout the text, as it refers to different intensities.

Done. 

Line 7-9. Please rephrase the sentence or watch the punctuation, as it is difficult to follow.

Done.

Line 13. Please change females for women throughout the text.

Done.

Line 19-20. Compare to what? To first trimester? Or to other types of activities? Please specify in abstract. Also, the second trimester does not have the highest results for occupational activities, if you see the respective table.

Corrections were done. Thank you. 

Introducition

Page 9 line 41. Change benefice for benefits.

Done.

Line 41-44. Could you cite any of those questionnaires that are too long or subjective?

Done. 

How is PPAQ not subjective, as it is a self-administred questionnaire. Please rewrite if necessary.

Corrections were done. This was added in the limitation section too.

Line 51. Add to the sentence that is the amount of time spent in the specific activity.

This is the sentence in line 51: “A valid PA tool in Arabic is necessary to help researchers assess, monitor and evaluate PA in North African and Middle Eastern pregnant females”.

We added as suggested: A valid PA tool in Arabic is necessary to help researchers assess, monitor and evaluate PA as well as the amount of time spent in each specific activity, in North African and Middle Eastern pregnant females.

Is this what you proposed?

Line 51. Start the sentence with Six instead of 6.

There is no 6 in the line 51.

Methods

Line 100. Don’t know if I understand well. The 45 pregnant females were all part of the university staff? If not, please, rephrase.

The participants of the pilot phase of the study were from our university community (lecturers, employees, students) and the affiliated hospital staff (nurses, employees, dietitians, doctors…).

Line 109. Mentioned where? In the introduction you say PA is important together with nutrition, but no mentioning of using another questionnaire. 

Data in relation with the nutritional status and physical activity will be presented in another article.

Furthermore, even if this particular study does not need that specific information as you used another complimentary questionnaire to assess that, maybe another study will use the full information, and therefore, the final Arabic version should take into account all the questions.

Done. We can add to the translated PPAQ to be shared in the future with other researchers those questions related to the date of recruitment, last menstrual period, and the predicted date of delivery.

Line 118. There are no other activities more usual in the Middle East that could be changed for those activities with a similar MET expenditure? This way the totals could be comparable with studies from other countries using the PPAQ.

In Middle Eastern and Arabic countries, religious constraints, lack of green spaces, and societal traditions don’t offer new choices or behaviors of physical activity.

Line 120. Please change “our” environment for “Middle Eastern” or “Arabic” countries.

Done

Line 127. Please cite the sutyd of there is a clinical trial or pprevious work published.

After reviewing publications on Pubmed and to our knowledge, our study is the first to evaluate physical activity level among women in Arab countries and the Middle-East. Our research team conducted in parallel, another study, with different objectives, among a different sample of women, and used our translated version of the PPAQ. We provided in our manuscript, some details related to this study and cited the proper reference.

Line 129. Singleton baby or singleton pregnancy?

Singleton pregnancy. It was corrected in the text. Thank you.

Line 134. Please refer to self-administer the newly translated PPAQ

It was mentioned in line 144 as follow:

“The last part of the questionnaire, concerned the newly translated PPAQ and was self-filled by the participants themselves”

Line 154. Please specify that the final number was 2 less in your study and you took that into account in the corresponding kind of activity/intensity totals.

We did not understand the comment. Line 154 is a subtitle. What do you suggest? If you can explain further please.

Statistical Analyses. Please indicate the corresponding table of results to each of the statistical analyses described.

The authors don’t specify any confounders for the ANOVA analyses comparing the PPAQ results by groups. However, when comparing, for example the PA levels by high education of low education (table 2) they don’t take into account how many women there are of each trimester of gestation, in each group. Even more, given the information that the PA levels differ significantly during the different trimesters, they may consider it as confounder. The same would apply for other tables and confounders.

In order to control confounding factors, univariates analyses followed by multivariate analyses were performed to assess the association between each explanatory variable and the PPAQ. A new table was added at the end of the result section, as follows.

Table 7. Multiple regression analysis of explanatory factors associated with PPAC in women

 Unstandardized Coefficients Standardized Coefficients t Sig. 95.0% Confidence Interval for B

 B Std. Error Beta Lower Bound Upper Bound

(Constant) 176.566 71.302 2.476 .014 35.831 317.300

Maternal age 1.654 1.281 .096 1.291 .198 -.875 4.183

Education 20.560 15.499 .099 1.327 .186 -10.031 51.150

Nutritional status [Ref: very bad/bad] -70.038 25.462 -.202 -2.751 .007 -120.294 -19.782

Physical status [Ref: very bad / Bad] 41.538 17.988 .171 2.309 .022 6.034 77.043

Results

162 add “of pregnancy” at the end of the sentence.

Done.

Line 204-205. Please keep the hypothesis and subjective approach for the discussion section with the proper citation.

These are lines 204-205:

“… sedentary lifestyle with a significant p value of 0.001 compared with females in their first or second trimester. In addition, household chores were significantly higher in women, in their second..”

in these lines, there is no hypothesis or subjective approach, they are results of the study that we present.

Tables. In all titles of the tables, please add “sedentary time” to the “Type and intensity of PA”. Also, please add units of measurement.

Done. Thank you.

Line 222. Add “respectively” after the P values.

Done.

Lines 224-228. Please put this information into the discussion section.

Line 224-228 are in fact the table 4 and not a text. So we did not understand the comment.

Discussion.

Line 251. The authors state “PA increased during the second trimester in our sample. Indeed, this period is considered to be the most comfortable period during gestation, where women often feel themselves more energetic and eager to be physically active”, while in the table, the total score for PPAQ is higher in the third trimester than the second, as well as in some particular intensities and activities. Also, please use references when explaining your results.

All this section was reviewed and corrected. New references were added.

Line 253-256. Please rephrase into shorter sentences (there is twice the word “since”)

The sentence was reformulated and corrected accordingly. Thank you.

277. Highlights.

Done

286-287. Final part of the sentence is not clear.

Corrections were done. Thank you.

294. How is the sample homogenous if there are women of all gestational ages and from different studies and economic backgrounds?

The sentence was reformulated and the word “homogenous” was deleted. Thank you.

Conclusions.

Line 303. Please rephrase the last sentence.

Done.

Line 310. Women

Done.
---

## [Editor Report · Decision Letter 2]

2 Mar 2020

Pregnancy physical activity questionnaire (PPAQ): translation and cross cultural adaption of an Arabic version.

PONE-D-19-25377R2

Dear Dr. Papazian,

We are pleased to inform you that your manuscript has been judged scientifically suitable for publication and will be formally accepted for publication once it complies with all outstanding technical requirements.

With kind regards,

Samson Gebremedhin, PhD

Academic Editor

PLOS ONE
---

## [Editor Report · Acceptance letter]

12 Mar 2020

PONE-D-19-25377R2 

Pregnancy physical activity questionnaire (PPAQ): translation and cross cultural adaption of an Arabic version. 

Dear Dr. Papazian:

I am pleased to inform you that your manuscript has been deemed suitable for publication in PLOS ONE. Congratulations! Your manuscript is now with our production department. 

With kind regards,

on behalf of

Dr. Samson Gebremedhin 

Academic Editor

PLOS ONE